# SIRT7 Deficiency Protects against Aging-Associated Glucose Intolerance and Extends Lifespan in Male Mice

**DOI:** 10.3390/cells11223609

**Published:** 2022-11-15

**Authors:** Tomoya Mizumoto, Tatsuya Yoshizawa, Yoshifumi Sato, Takaaki Ito, Tomonori Tsuyama, Akiko Satoh, Satoshi Araki, Kenichi Tsujita, Masaru Tamura, Yuichi Oike, Kazuya Yamagata

**Affiliations:** 1Department of Medical Biochemistry, Faculty of Life Sciences, Kumamoto University, Kumamoto 860-8556, Japan; 2Department of Medical Technology, Faculty of Health Science, Kumamoto Health Science University, Kumamoto 861-5598, Japan; 3Center for Metabolic Regulation of Healthy Aging, Faculty of Life Sciences, Kumamoto University, Kumamoto 860-8556, Japan; 4Department of Integrative Physiology, National Center for Geriatrics and Gerontology, Obu 474-8511, Japan; 5Department of Integrative Physiology, Institute of Development, Aging, and Cancer, Tohoku University, Sendai 980-8575, Japan; 6Department of Cardiovascular Medicine, Faculty of Life Sciences, Kumamoto University, Kumamoto 860-8556, Japan; 7Technology and Development Team for Mouse Phenotype Analysis, RIKEN BioResource Research Center, Tsukuba-shi 305-0074, Japan; 8Department of Molecular Genetics, Faculty of Life Sciences, Kumamoto University, Kumamoto 860-8556, Japan

**Keywords:** SIRT7, knockout mouse, lifespan, FGF21

## Abstract

Sirtuins (SIRT1–7 in mammals) are evolutionarily conserved nicotinamide adenine dinucleotide-dependent lysine deacetylases/deacylases that regulate fundamental biological processes including aging. In this study, we reveal that male *Sirt7* knockout (KO) mice exhibited an extension of mean and maximum lifespan and a delay in the age-associated mortality rate. In addition, aged male *Sirt7* KO mice displayed better glucose tolerance with improved insulin sensitivity compared with wild-type (WT) mice. Fibroblast growth factor 21 (FGF21) enhances insulin sensitivity and extends lifespan when it is overexpressed. Serum levels of FGF21 were markedly decreased with aging in WT mice. In contrast, this decrease was suppressed in *Sirt7* KO mice, and the serum FGF21 levels of aged male *Sirt7* KO mice were higher than those of WT mice. Activating transcription factor 4 (ATF4) stimulates *Fgf21* transcription, and the hepatic levels of *Atf4* mRNA were increased in aged male *Sirt7* KO mice compared with WT mice. Our findings indicate that the loss of SIRT7 extends lifespan and improves glucose metabolism in male mice. High serum FGF21 levels might be involved in the beneficial effect of SIRT7 deficiency.

## 1. Introduction

Sirtuins (SIRT1–7 in mammals) are evolutionarily conserved nicotinamide adenine dinucleotide-dependent lysine deacetylases/deacylases that regulate diverse biological processes, including metabolism, stress responses, genomic stability, and aging [1]. Genetic overexpression of sirtuins increases longevity in a variety of lower organisms such as yeast, worms, and flies [2,3,4]. Intriguingly, transgenic mice with brain-specific *Sirt1* overexpression and whole-body *Sirt6* transgenic mice also show an extended lifespan [5,6,7].

Metabolic dysfunction, including increased body fat and reduced glucose tolerance, is a hallmark of aging [8]. SIRT1, SIRT6, and SIRT7 are nuclear proteins, and SIRT1 and SIRT6 exert beneficial effects against metabolic diseases [9,10]. In sharp contrast, we demonstrated that *Sirt7* knockout (KO) mice are resistant to high-fat diet-induced obesity, glucose intolerance, and fatty liver [11], suggesting that the loss of SIRT7 induces a metabolically healthy condition. Aging is also a major risk factor for cancer. With regard to cancer, SIRT7 is responsible for tumor phenotype maintenance by deacetylation of histone H3 lysine 18 (H3K18) [12], and SIRT7 expression is upregulated in the majority of human cancers, including hepatic, gastric, colorectal, and breast cancers [13,14]. In addition, SIRT7 may exert its oncogenic properties through the upregulation of ribosomal RNA synthesis to meet the increased demand for ribosomes in rapidly growing tumor cells [13,15,16]. Considering that SIRT1 and SIRT6 act as tumor suppressors [17,18], SIRT7 seems to have the opposite role in cancer. Of note, ribosomal protein gene deletion and inhibition of translation have been reported to extend lifespan in numerous model organisms, including mammals [12,19,20]. Together, these findings suggest the possibility that SIRT7 deficiency plays beneficial roles in aging-associated metabolic disorders, cancer, and even lifespan. However, a previous study reported that *Sirt7* KO mice exhibit a shortened lifespan with severe cardiac disorders [21].

Because our *Sirt7* KO mice [11] did not display such a shortened lifespan, in the present study, we reevaluated the impact of the loss of SIRT7 on lifespan in mice. We found that male, but not female, *Sirt7* KO mice on a C57BL/6J background showed an extension of mean and maximum lifespan and a delay of the age-associated mortality rate. In addition, aged male *Sirt7* KO mice displayed better glucose tolerance and higher serum levels of fibroblast growth factor 21 (FGF21) compared with wild-type (WT) mice. It has been reported that FGF21 improves glucose metabolism and extends lifespan in mice [22,23]; therefore, increased levels of FGF21 might be involved in the improved glucose tolerance and lifespan extension of male *Sirt7* KO mice.

## 2. Materials and Methods

### 2.1. Mice

*Sirt7* KO mice were obtained from Dr. Eva Bober [21]. These mice were backcrossed to C57BL/6J mice (Charles River Laboratory Japan, Inc., Kanagawa, Japan) for at least five generations. Male and female *Sirt7* heterozygous mice were crossed to obtain WT and *Sirt7* KO littermates. Genomic DNA was isolated from a 3-week-old mouse tail and PCR genotyping was performed as previously described [11]. All mouse experiments were performed in accordance with the guidelines of the Institutional Animal Committee of Kumamoto University. The mice were housed at a maximum of 5 mice/cage and maintained at 22 ± 2 °C with a 12-h light/dark cycle and free access to water and normal chow (CE-2; CLEA Japan, Inc., Tokyo, Japan).

### 2.2. Echocardiography and Cardiac Hypertrophy Measurements

Thirty-month-old male WT and *Sirt7* KO mice were lightly anesthetized with 1% isoflurane for shaving and quickly subjected to echocardiography. Transthoracic echocardiography was performed using an Aprio 300 (Toshiba Corp., Tokyo, Japan) in awake and conscious conditions. For echocardiography, we held the mice gently with their back toward the palm and placed the transducer on the chest while avoiding the vagal reflex induced by the pressure of the transducer [24]. Left ventricular wall thickness, left ventricular end-diastolic dimension, left ventricular end-systolic dimension, and percentage fractional shortening were calculated in M-mode. To quantify cardiac hypertrophy, mouse hearts were dissected immediately, and heart weight and tibia length were measured to calculate the heart weight/tibia length ratio. Cross-sectional images of hematoxylin and eosin-stained cardiomyocytes were captured using a BZ-710 All-in-One Microscope (Keyence, Inc., Osaka, Japan), and the cross-sectional area of 50–100 cardiomyocytes in each section was measured with ImageJ software.

### 2.3. Lifespan Study

Male (WT, *n* = 50, *Sirt7* KO, *n* = 42) and female (WT, *n* = 40, *Sirt7* KO, *n* = 34) mice were inspected at least twice a day for health issues and their age was recorded when the mice were found dead. Animals showing signs of morbidity (immobility, lack of responsiveness to manual stimulation, and inability to eat or drink) were euthanized by manual cervical dislocation according to the institutional animal care guidelines of Kumamoto University. The time at euthanization was the endpoint. Lifespan was assessed using Kaplan–Meier survival curves. To estimate the hazard ratio, 2*qx*/(2 − *qx*), the age-related mortality rate (*qx*) was estimated as the number of mice at the end of an interval against the number of mice at the beginning of the interval [25]. The natural logarithm of the hazard ratio was plotted. Mice used in the longevity study were not used for any other experiment.

### 2.4. Histological Analyses

Mouse tissues (brain, thyroid, trachea, lung, stomach, pancreas, liver, spleen, intestine, kidney, urinary bladder, and skeletal muscle) were recovered from male WT (*n* = 22) and *Sirt7* KO mice (*n* = 21) soon after death. The tissues were fixed in phosphate-buffered 4% paraformaldehyde and embedded in paraffin; 4-μm-thick sections were obtained and stained with hematoxylin and eosin. The sections were examined for neoplasms by a pathologist (T. I.).

The heart sections from each mouse group were stained with Masson’s trichrome stain to detect fibrosis.

### 2.5. Serum Parameters and Enzyme-Linked Immunosorbent Assay Measurements

After 16-h fasting, blood samples were collected from 24-month-old male WT and *Sirt7* KO mice by cardiac puncture. Serum preparation was performed by centrifugation at 5000× *g* for 10 min at 4 °C using a blood collection tube (TS-801; SATO Chemical Industry Co., Ltd., Tochigi, Japan). Biochemical parameters were measured using an automatic biochemical analyzer (JCA-BM6070; JEOL Ltd., Tokyo, Japan). Serum hormone levels (except for adiponectin) were measured by using a Bio-Plex Pro Mouse Diabetes 8-Plex panel (Bio-Rad Laboratories, Inc., Hercules, CA, USA), a MILLIPLEX Mouse Myokine Magnetic Bead Panel (Millipore Co., Bedford, MA, USA), and a Bio-Plex200 system (Bio-Rad Laboratories, Inc.). Adiponectin was measured using a Bio-Plex Pro Mouse Adiponectin Assay Kit (171F7002M; Bio-Rad Laboratories, Inc.).

For serum FGF21 measurements (for Figure 5), blood was collected from the heart after 16-h fasting and centrifuged at 5000× *g* for 10 min at 4 °C to obtain serum. Serum samples from 4- and 30-month-old male mice were assayed for the quantification of FGF21 using an enzyme-linked immunosorbent assay kit (MF2100; R&D Systems, Inc., Minneapolis, MN, USA). An iMark™ Microplate Reader (Bio-Rad Laboratories, Inc.) was used to read samples at 450 nm and corrected at 540 nm.

### 2.6. Metabolic Tests

The body weight of 24-month-old male WT and *Sirt7* KO mice was measured. For the glucose tolerance test, the mice were injected intraperitoneally with 2 g/kg glucose after 15-h fasting. Blood was collected from the tail vein and blood glucose was monitored by a Glutest Neo Super (Sanwa Kagaku Kenkyusyo Co., Ltd., Nagoya, Japan) at the indicated time points (0, 15, 30, 60, and 120 min). For the insulin tolerance test, 1 U/kg insulin was injected intraperitoneally after 4-h fasting. Blood insulin was monitored at the indicated time points (0, 30, 60, 90, and 120 min).

### 2.7. RNA-seq Analysis

Livers were collected from 24-month-old male WT and *Sirt7* KO mice after 16-h fasting, and RNA isolation was performed using a ReliaPrep RNA Miniprep System (Z6012; Promega, Inc., Madison, WI, USA). RNA quality was determined with the RNA integrity number equivalent value using a High Sensitivity RNA ScreenTape Assay (5067-5579; Agilent Technologies, Inc., Santa Clara, CA, USA), and confirmed to be 6.9 or higher. Total RNA (1 ng) was used for the reverse transcription reaction with a SMART-seq HT (634455; Takara Bio, Inc., Shiga, Japan). An RNA-seq library was prepared using a Nextera XT Library Prep Kit (FC-131-1024; Illumina, Inc., San Diego, CA, USA) and sequenced on a NextSeq 500 Sequencer with 75 bp single-end reads (Illumina). The reads were trimmed for universal Illumina adaptors with Trim Galore (version 0.6.5) [26] and mapped to the mouse transcriptome (GRCm38) and quantified by Salmon (version 1.2.1) with default settings. Differential expression testing was performed with DESeq2 (version 1.28.0). Data were loaded into R using tximport (version 1.16.0) and aggregated to gene-level abundance in TPM. Differentially expressed genes were defined as *p* < 0.05, fold change > 1.5. Upregulated differentially expressed genes in *Sirt7* KO mice were subjected to enrichment analysis with DAVID (version 6.8) [27]. Gene Ontology and Kyoto Encyclopedia of Genes and Genomes pathway analyses were used for annotation.

### 2.8. qRT-PCR

Frozen tissues were homogenized in Sepasol-RNA I Super G Solution (Nacalai Tesque, Inc., Kyoto, Japan). Total RNA was extracted using the phenol-chloroform extraction method, and cDNA was synthesized from total RNA using a Prime Script RT Kit (RR047A; Takara). qRT-PCR was performed on an ABI 7300 thermal cycler (Applied Biosystems, Foster City, CA, USA) using SYBR Premix Ex Taq II (RR820A; Takara). Relative gene expression was normalized by the mRNA expression level of mouse TATA-binding protein. The primer sequences are shown in Appendix A.

### 2.9. Statistical Analysis

Data are presented as the mean ± standard deviation (SD). The significance of differences was calculated with an unpaired two-tailed Student’s *t*-test or two-way analysis of variance with Tukey’s post hoc test. Kaplan–Meier survival curves were compared using the log-rank test. Changes in the age-associated mortality rate (slope and *y*-intercept) were measured using analysis of covariance. The frequency of cancer was compared using Fisher’s exact test. Statistical analysis was performed using GraphPad Prism 9 software version 9.4.0 (GraphPad Software, Inc., San Diego, CA, USA). Statistical significance was assumed at *p* < 0.05.

## 3. Results

### 3.1. Lack of Cardiac Dysfunction in Aged Sirt7 KO Mice

Vakhrusheva et al. [21] reported that *Sirt7* KO mice (C57BL/6 × 129Sv mixed background mice were backcrossed to C57BL/6) exhibit inflammatory cardiomyopathy with a strong increase in fibrosis and shortened lifespan. Therefore, we investigated the cardiac function of our 30-month-old male *Sirt7* KO mice on a C57BL/6J background by echocardiographic analysis. Interestingly, there were no significant differences in left ventricular dimension and contractile function between WT and *Sirt7* KO mice (Figure 1A,B). The heart weight to tibial length ratio and lung weight to tibial length ratio of *Sirt7* KO mice were both similar to those of WT mice (Figure 1C). Histological analysis revealed that the cross-sectional area, showing relative cardiomyocyte size, was similar between WT and *Sirt7* KO mice (Figure 1D,E). Fibrosis was not detected in the heart of our *Sirt7* KO mice by Masson’s trichrome staining (Figure 1F). Consistently, the cardiac mRNA expression levels of *Col3a1* (encoding collagen type III) and *Col6a1* (encoding collagen type VI) were similar between WT and *Sirt7* KO mice (Figure 1G). In addition, increased expression of inflammatory genes was not detected in the heart of *Sirt7* KO mice (Figure 1H). Taken together, these observations show that our aged *Sirt7* KO mice do not exhibit cardiac dysfunction compared with WT mice.

### 3.2. Male Sirt7 KO Mice Exhibit an Extension of Lifespan

These findings encouraged us to reevaluate the lifespan of *Sirt7* KO mice. When fed a standard chow diet, neither male nor female *Sirt7* KO mice on a C57BL/6J background exhibited a shortened lifespan (Figure 2A). Interestingly, the survival curves of WT and *Sirt7* KO male, but not female, mice were significantly different by log-rank testing (*p* = 0.0053), and male *Sirt7* KO mice showed an 11.2% extension of median lifespan (the day at which the probability of survival equals 50%: WT 887 days vs. *Sirt7* KO 972 days) (Figure 2A,B). They also exhibited a significant extension of mean lifespan (WT 874 ± 149 days vs. *Sirt7* KO 939 ± 130 days, *p* = 0.031). Furthermore, the maximum lifespan (the average of the mean lifespan of the longest-lived 10% or 20% of mice) [6,28] of male *Sirt7* KO mice was also significantly increased (10% oldest WT 1053 ± 19 days vs. *Sirt7* KO 1086 ± 4 days, *p* = 0.021; 20% oldest WT 1038 ± 21 days vs. *Sirt7* KO 1076 ± 10 days, *p* = 0.0005) (Figure 2B). We next assessed age-associated mortality in *Sirt7* KO mice. Male *Sirt7* KO mice exhibited a significant delay in age-associated mortality compared with WT mice, whereas the slope of age-associated mortality change, which defines the rate of aging [25], did not differ between WT and *Sirt7* KO mice (Figure 2C). In contrast to males, this delay in age-associated mortality was not detected in female *Sirt7* KO mice. Taken together, these results suggest that SIRT7 deficiency extends lifespan in male mice by delaying the onset of age-associated physiological decline.

Malignant neoplasm is a major cause of death in laboratory mice. Because SIRT7 has oncogenic properties [13,14], we investigated whether the extension of lifespan in male *Sirt7* KO mice was due to the reduced incidence of neoplasms. However, post-mortem gross and microscopic examinations revealed that the incidence of malignant neoplasms was similar between male WT (11 out of 22 mice, 50%) and *Sirt7* KO mice (12 out of 21 mice, 57.1%) (Figure 3A). In addition, the average number of tumors per mouse did not differ between WT and *Sirt7* KO mice (Figure 3B). These results indicate that the pro-longevity effect of SIRT7 deficiency cannot be reasoned by the decreased incidence of neoplasms.

### 3.3. Sirt7 KO Mice are Protected from Aging-Associated Metabolic Dysfunction

Next, we examined serum biochemical parameters in fasted 24-month-old male mice. Serum uric acid levels and high-density lipoprotein (HDL) levels were significantly lower and higher, respectively, in aged *Sirt7* KO mice than in WT mice (Figure 4A). High HDL levels are associated with a reduced risk of atherosclerosis and cardiovascular diseases in humans [29], but the significance of increased HDL levels on lifespan extension in mice is unclear. There was a trend for altered levels of alanine aminotransferase, total bilirubin, and lactate dehydrogenase in aged *Sirt7* KO mice, but the difference did not reach significance probably due to small sample sizes. Metabolic dysfunction is a hallmark of aging. As glucose tolerance and insulin sensitivity were improved in *Sirt7* KO mice fed a high-fat diet [11], we next investigated the metabolic parameters of aged male mice. *Sirt7* KO animals weighed significantly less than WT mice (Figure 4B). A glucose tolerance test demonstrated better glucose tolerance in aged *Sirt7* KO mice (Figure 4C), and an insulin tolerance test revealed significantly better insulin tolerance in aged *Sirt7* KO mice (Figure 4D). These results indicate that the loss of SIRT7 confers protection against aging-associated dysfunction in glucose metabolism.

### 3.4. Hepatic FGF21 Expression is Maintained in Aged Sirt7 KO Mice

We next measured the serum levels of secreted polypeptides and proteins that are involved in glucose metabolism. Among the secreted factors, the serum levels of FGF21 were significantly increased in 24-month-old male *Sirt7* KO mice compared with WT mice (Table 1). FGF21 has a fundamental role in the regulation of energy expenditure, and FGF21 administration promotes weight loss and improves insulin sensitivity and glucose homeostasis [22]. Moreover, transgenic mice overexpressing FGF21 show an extended lifespan [22,30,31,32]. We also measured serum FGF21 levels in male mice in an independent cohort (Figure 5A). The serum FGF21 levels of young mice did not differ between genotypes. Serum FGF21 levels were markedly decreased with aging in WT mice. In contrast, this decrease was suppressed in *Sirt7* KO mice, and the increased serum FGF21 levels of aged male *Sirt7* KO mice were confirmed in this cohort.

The liver is a major source of the expression and secretion of FGF21 [33]. Hepatic *Fgf21* mRNA expression levels were similar between young male *Sirt7* KO and WT mice (Figure 5B). In accordance with a previous report [34], hepatic *Fgf21* mRNA expression decreased markedly with aging in male WT mice, but such a decrease was not detected in *Sirt7* KO mice, and *Fgf21* mRNA expression was significantly increased in aged male *Sirt7* KO mice compared with WT mice (Figure 5B). FGF21 induces the expression of genes involved in glucose transport (*Slc2a1* and *Slc2a4*), mitochondrial oxidation (*Cycs*), and thermogenesis (*Ucp1*, *Dio2*, and *Ppargc1a*) in brown adipose tissue and lipid metabolism (*Lipe* and *Pnpla2*) in white adipose tissue [35,36]. The expression of these genes was significantly increased in aged male *Sirt7* KO mice compared with WT mice (Figure 5C,D). These results indicate that the increased serum levels of FGF21 in aged male *Sirt7* KO mice are not due to FGF21 resistance, a state of impaired FGF21 signaling [37].

To further investigate the mechanism underlying the increase in hepatic FGF21 expression in aged male *Sirt7* KO mice, we performed RNA-seq analysis of the liver. We identified 243 differentially expressed genes (173 upregulated and 70 downregulated; adjusted *p* < 0.05) in the liver of aged male *Sirt7* KO mice compared with WT mice (Figure 5E). Intriguingly, Gene Ontology and Kyoto Encyclopedia of Genes and Genomes analyses revealed that genes involved in the unfolded protein response of the endoplasmic reticulum (UPR^ER^), including *Hspa5* (encoding GRP78) and *Pdia3* (encoding protein disulfide isomerase, family A, member 3), were increased in the liver of aged male *Sirt7* KO mice (Figure 5E). Altered ER homeostasis leads to the accumulation of unfolded proteins in the ER, called ER stress, which activates the UPR^ER^ signaling pathway to mitigate the stress [38,39]. Activating transcription factor 4 (ATF4) is one of the main effectors of the UPR^ER^ and controls the expression of stress-resistance genes including *Fgf21* [22]. Therefore, we analyzed the hepatic expression of genes involved in the UPR^ER^ in male WT and *Sirt7* KO mice by qRT-PCR (Figure 5F). The hepatic expression of several UPR^ER^-related genes such as *Atf4* and *Hspa5* decreased with aging in male WT mice, whereas this decrease was diminished in *Sirt7* KO mice. As a result, *Atf4* mRNA expression was significantly increased in aged *Sirt7* KO mice compared with WT mice. ATF4 can induce apoptosis via the induction of *Ddit3* (encoding C/EBP-homologous protein); however, *Ddit3* expression was similar between male *Sirt7* KO and WT mice (Figure 5F). *Cdkn2a* (encoding p16, a marker of senescence) expression did not differ between aged male *Sirt7* KO and WT mice (Figure 5G), indicating that cellular senescence occurs similarly in the hepatic cells of these mice. These results suggest that the loss of SIRT7 helps to maintain serum FGF21 levels at high levels in aged male *Sirt7* KO mice by maintaining hepatic ATF4 expression.

## 4. Discussion

The phenotypes of *Sirt7* KO mice are controversial. Vazquez et al. reported that their *Sirt7* KO mice exhibit increased perinatal lethality [40], but such an increase was not observed in our *Sirt7* KO mice [41] or in another independent line of *Sirt7* KO mice [42]. We have no adequate explanation for these contrasting results, but differences in the construct used (absence of LacZ in [11] and [42]) may have contributed to these discrepancies. A reduction in SIRT7 levels with aging has been reported in tissues of humans and animal models [43], while SIRT7 levels increase during calorie restriction [44,45], suggesting that SIRT7 might exhibit a protective effect on longevity. Accordingly, Vakhrusheva et al. [21] reported that their *Sirt7* KO mice exhibit a shortened lifespan due to cardiac dysfunction with a strong increase in fibrosis [21]. Although we and Vakhrusheva et al. [21] used the same *Sirt7* KO line, such phenotypes were not detected in this study. The reason is again unclear, but the lack of cardiac dysfunction may have contributed to the extended lifespan of our *Sirt7* KO mice. We also would like to emphasize that the phenotype of shortened lifespan was not observed in another line of *Sirt7* KO mice [42] (Professor Johan Auwerx, personal communication). We backcrossed Vakhrusheva’s *Sirt7* KO mice (C57BL/6 × 129Sv mixed background was backcrossed onto C57BL/6 background) with C57BL/6J mice for at least five generations before use in this study. Ryu et al. backcrossed their *Sirt*7 KO mice (129Sv background) for ten generations onto the C57BL/6J background [42]. Since genetic background affects lifespan in mice [28], the different genetic backgrounds of these mice might have contributed to their altered cardiac phenotypes and lifespans. Furthermore, differences in environmental factors (e.g., diet and housing conditions) may have affected their phenotypes. Further studies are necessary to clarify the reasons for these discrepancies.

FGF21 is an endocrine hormone that exerts a fundamental role in the regulation of energy metabolism, and FGF21 administration promotes weight loss and improves glucose homeostasis by enhancing insulin sensitivity [22]. In addition, FGF21 is a potent longevity factor, and transgenic mice overexpressing FGF21 show an extended lifespan [30]. We found that serum FGF21 levels were significantly higher in our aged male *Sirt7* KO mice than in aged male WT mice. Therefore, there is a possibility that the increase in serum FGF21 levels might contribute to the extended lifespan of male *Sirt7* KO mice. However, female *Sirt7* KO mice did not show an extension of lifespan, but we did not examine their serum FGF21 levels. Thus, we cannot conclude that the prolonged lifespan of our male *Sirt7* KO mice was due to increased FGF21 levels. It has been reported that female mice exhibit higher serum concentrations of FGF21 than males [46]. Investigation of serum FGF21 levels in female mice is a major future undertaking to address the contribution of FGF21 to the prolonged lifespan of male *Sirt7* KO mice.

We also revealed that aged male *Sirt7* KO mice showed less body weight gain and improved glucose homeostasis compared with WT mice of the same age. FGF21 promotes weight loss and enhances insulin sensitivity by increasing energy expenditure [22]. Hence, it is plausible that the increased levels of FGF21 contribute to the improved glucose metabolism in aged male *Sirt7* KO mice. Additionally, FGF21 is reported to increase serum HDL-cholesterol levels [47]. Thus, the increased serum HDL-cholesterol concentrations in aged male *Sirt7* KO mice might also be attributable to the upregulation of FGF21. However, we do not claim that the improved glucose and lipid metabolism in male *Sirt7* KO mice can be explained by the increase in FGF21 only. Serum FGF21 levels were similar between young *Sirt7* KO and control mice, but we demonstrated previously that young *Sirt7* KO mice show resistance to high-fat diet-induced obesity, glucose intolerance, and fatty liver [11].

The diminished ability to maintain protein homeostasis is a hallmark of aging, and aged cells are unable to properly trigger UPR^ER^-related transcriptional responses [48,49,50]. Consistently, we demonstrated that the expression of several hepatic UPR^ER^-related genes, including *Atf4*, was decreased in aged male WT mice. In sharp contrast, this decrease was suppressed in aged male *Sirt7* KO mice, and *Atf4* mRNA expression was significantly increased in aged male *Sirt7* KO mice compared with WT mice. Given that ATF4 stimulates *Fgf21* transcription, the increased hepatic *Fgf21* expression and serum FGF21 levels in aged male *Sirt7* KO mice might be, at least in part, a consequence of higher ATF4 expression. However, it is unlikely that *Atf4* expression is regulated directly by SIRT7 in the liver since the hepatic *Atf4* mRNA levels of young mice did not differ between genotypes. What could be the underlying mechanism for the altered expression of UPR^ER^-related genes in aged male *Sirt7* KO mice? Previous studies have demonstrated that the chromatin landscape of aged cells is largely different from that of young cells, and the reduction of activating histone marks and the induction of repressive marks at the promoter regions of stress response genes are features of aged chromatin [48,49]. H3K18 acetylation, H3K36 acetylation, and H4K91 glutarylation are histone marks for active gene expression, and SIRT7 functions as an eraser of these modifications [12,51,52]. Loss of SIRT7 may help to induce gene expression by preserving active histone marks. Future studies are necessary to define whether histone marks at UPR^ER^-related gene loci are regulated by SIRT7.

Enhancing the function of SIRT1 and SIRT6 improves the healthspan in mice. Although the phenotypes of *Sirt7* KO mice are controversial, we revealed that the loss of SIRT7 extends lifespan and confers protection against aging-associated metabolic dysfunction in male mice. Thus, SIRT7 and SIRT1/SIRT6 may play opposite roles in aging. Further studies are necessary to improve our understanding of the roles of SIRT7 in aging.

## Figures and Tables

**Figure 1 cells-11-03609-f001:**
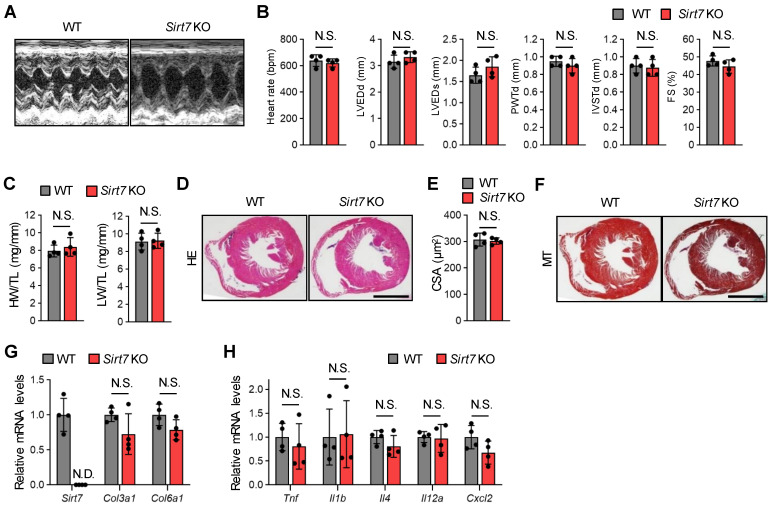
Cardiac function and morphology in aged male *Sirt7* KO mice. (**A**) Representative M-mode echocardiogram images of 30-month-old male WT and *Sirt7* KO mice. (**B**) Quantitative analysis of heart rate, left ventricular end-diastolic diameter (LVEDd), left ventricular end-systolic diameter (LVED), posterior wall thickness in diastole (PWTd), interventricular septum thickness in diastole (IVSTd), and fractional shortening (FS) in 30-month-old male WT and *Sirt7* KO mice (*n* = 4). (**C**) Quantification of the heart weight/tibia length (HW/TL) ratio and lung weight/tibia length (LW/TL) ratio in 30-month-old male WT and *Sirt7* KO mice (*n* = 4). (**D**) Representative images of hematoxylin and eosin (HE)-stained heart tissue from 30-month-old male WT and *Sirt7* KO mice. Scale bar, 2 mm. (**E**) Quantitative analysis of cardiomyocyte cross-sectional area in 30-month-old male WT and *Sirt7* KO mice (*n* = 4). (**F**) Representative images of Masson’s trichrome (MT)-stained heart tissue from 30-month-old male WT and *Sirt7* KO mice. Scale bar, 2 mm. (**G**,**H**) qRT-PCR analysis of *Sirt7* and fibrosis-related genes (*Col3a1* and *Col6a1*) (G) and inflammation-related genes (*Tnf*, *Il1b*, *Il4*, *Il12a*, and *Cxcl2*) (H) in the heart of 30-month-old male WT and *Sirt7* KO mice (*n* = 4). The data are expressed as the mean ± SD; N.D., not detected; N.S., not significant by unpaired Student’s *t*-test.

**Figure 2 cells-11-03609-f002:**
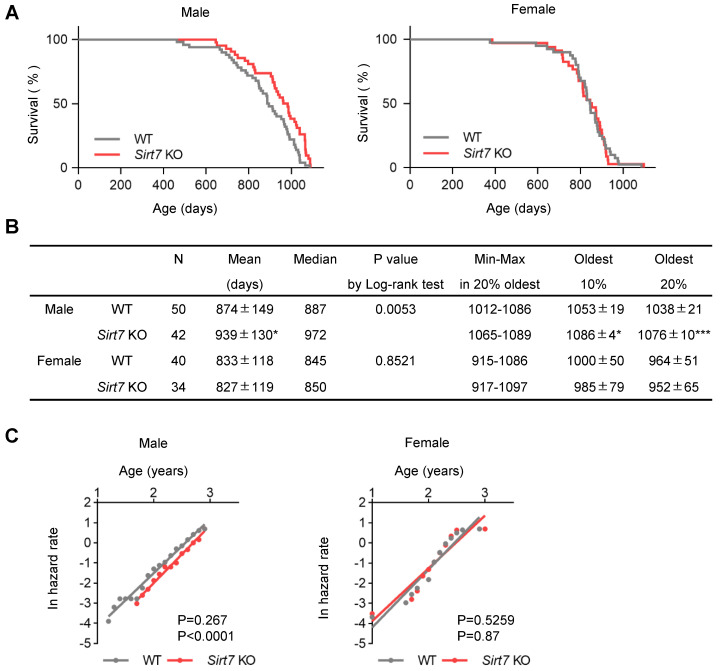
Lifespan analysis of male and female *Sirt7* KO mice. (**A**) Kaplan–Meier survival curves for male (left: WT [*n* = 50], *Sirt7* KO [*n* = 42]) and female (right: WT [*n* = 40], *Sirt7* KO [*n* = 34]) mice. (**B**) Parameters of lifespan analysis. The mean (average lifespan) and oldest 10% and 20% (mean lifespan of the longest-lived 10% and 20% mice) of each group are shown as the mean ± SD; * *p* < 0.05, *** *p* < 0.001 by unpaired Student’s *t*-test. *P*-values of (**A**) were calculated by the log-rank test. (**C**) The age-associated mortality rate of male (left) and female (right) WT and *Sirt7* KO mice. *P*-values for the differences between the slopes (age-associated mortality rate) and *y*-intercepts (initial mortality rate) were calculated by analysis of covariance (top = *p*-value of slopes, bottom = *p*-value of *y*-intercepts).

**Figure 3 cells-11-03609-f003:**
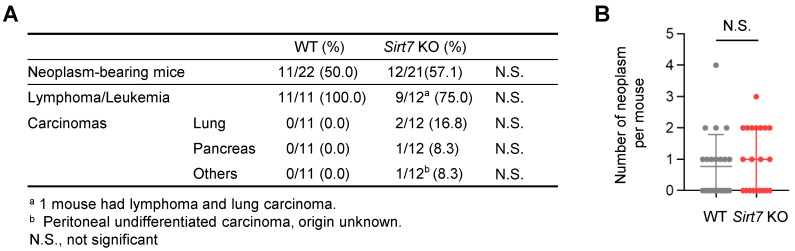
Tumor spectrum and incidence in aged male *Sirt7* KO mice. (**A**) Microscopic pathological findings at death in male WT (*n* = 22) and *Sirt7* KO (*n* = 21) mice. Statistically significant differences were calculated using Fisher’s exact test. (**B**) Number of neoplasms in male WT (*n* = 22) and *Sirt7* KO (*n* = 21) mice. Data are expressed as the mean ± SD; N.S., not significant by unpaired Student’s *t*-test.

**Figure 4 cells-11-03609-f004:**
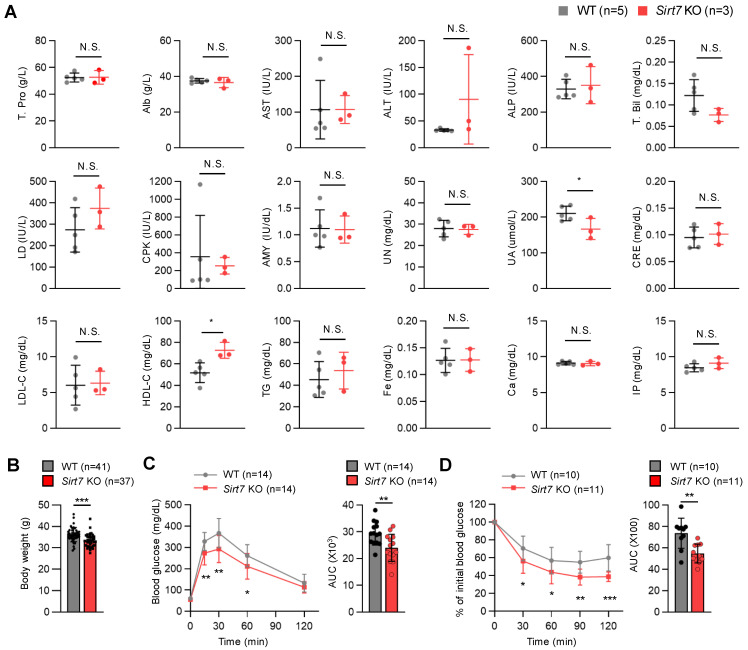
Serum biochemical parameters and glucose metabolism in aged male *Sirt7* KO mice. (**A**) Biochemical analysis of 24-month-old male WT (*n* = 5) and *Sirt7* KO (*n* = 3) mice after 16-h fasting. (**B**) Body weight of 24-month-old male WT (*n* = 41) and *Sirt7* KO (*n* = 37) mice. (**C**) Glucose tolerance test in 24-month-old male WT (*n* = 14) and *Sirt7* KO (*n* = 14) mice after intraperitoneal injection of glucose (2 g/kg body weight). (**D**) Insulin tolerance test in 24-month-old male WT (*n* = 10) and *Sirt7* KO (*n* = 11) mice after intraperitoneal injection of insulin (1 U/kg body weight). The area under the curve (AUC) for each tolerance test (**C**,**D**) is calculated and shown. Data are expressed as the mean ± SD; * *p* < 0.05, ** *p* < 0.01, *** *p* < 0.001; N.S., not significant by unpaired Student’s *t*-test. Alb, albumin; ALP, alkaline phosphatase; ALT, alanine aminotransferase; AMY, amylase; AST, aspartate aminotransferase; Ca, calcium; CPK, creatine phosphokinase; CRE, creatinine; Fe, iron; HDL-C, high-density lipoprotein cholesterol; IP, inorganic phosphorus; LD, lactate dehydrogenase; LDL-C, low-density lipoprotein cholesterol; T. Bil, total bilirubin; TG, triglyceride; T. Pro, total protein; UA, uric acid; UN, urea nitrogen.

**Figure 5 cells-11-03609-f005:**
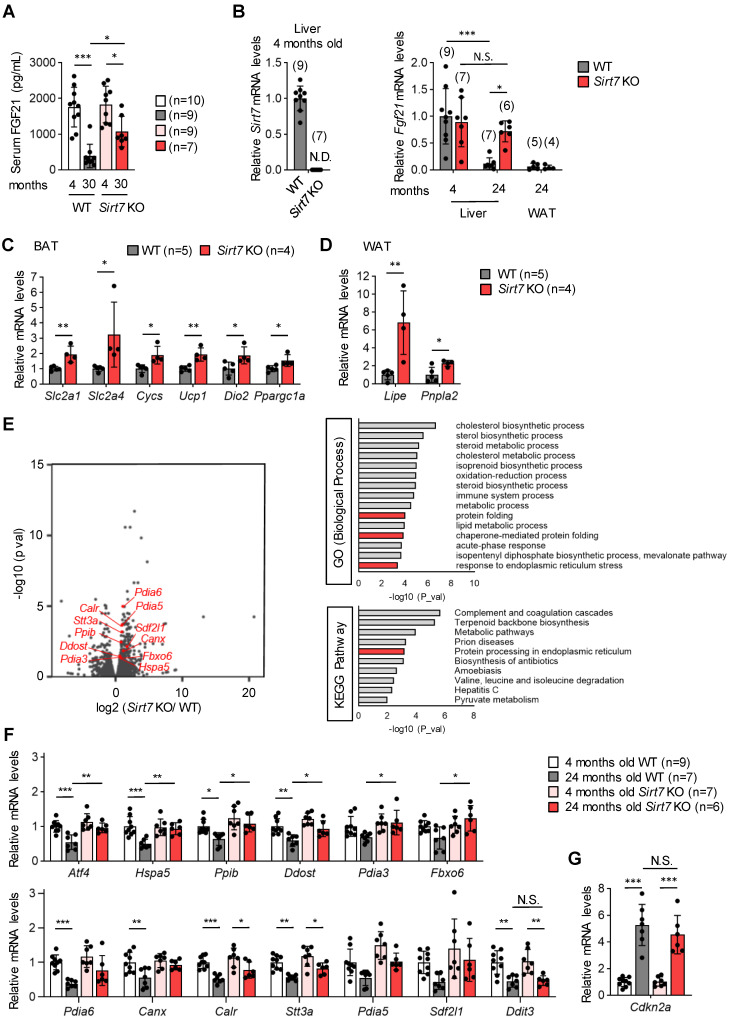
Gene expression profiles in the liver of aged male *Sirt7* KO mice. (**A**) Serum FGF21 levels in 4- or 30-month-old male WT and *Sirt7* KO mice after 16-h fasting. (**B**) qRT-PCR analysis of *Sirt7* and *Fgf21* in the liver and epididymal white adipose tissue (WAT) from male WT and *Sirt7* KO mice. N.D., not detected. (**C**,**D**) qRT-PCR analysis of *Fgf21* target genes in brown adipose tissue (BAT) (*Slc2a1*, *Slc2a4*, *Cycs*, *Ucp1*, *Dio2*, and *Ppargc1a*) (**C**) and epididymal WAT (*Lipe* and *Pnpla2*) (**D**) of 24-month-old male WT and *Sirt7* KO mice. (**E**) Volcano plot (left) and enrichment analysis (right) of RNA-seq data in liver samples from 24-month-old male WT and *Sirt7* KO mice (*n* = 3). UPR^ER^-related genes are shown in red (left). GO, Gene Ontology; KEGG, Kyoto Encyclopedia of Genes and Genomes. (**F**,**G**) qRT-PCR analysis of UPR^ER^-related genes (**F**) and a marker of senescence (**G**) in the liver of 4- and 24-month-old male WT and *Sirt7* KO mice. Data are expressed as the mean ± SD; * *p* < 0.05, ** *p* < 0.01, *** *p* < 0.001; N.S., not significant. Statistical significance was determined by either two-way analysis of variance with Tukey’s post hoc test (**A**,**B**,**F**,**G**) or unpaired Student’s *t*-test (**C**,**D**).

**Table 1 cells-11-03609-t001:** Serum hormone levels in aged male WT and *Sirt7* KO mice.

Parameter	WT (*n* = 5)	*Sirt7* KO (*n* = 3)	*p* Value
Insulin (ng/mL)	7.1 ± 5.2	4.9 ± 0.8	0.51
Glucagon (ng/mL)	10.3 ± 5.9	6.8 ± 1.2	0.37
GLP-1 (pg/mL)	789.6 ± 650.3	377.2 ± 138.4	0.33
GIP (ng/mL)	6.3 ± 1.0	6.2 ± 0.1	0.88
Ghrelin (ng/mL)	26.9 ± 6.6	24.0 ± 1.5	0.49
PAI-1 (ng/mL)	2.5 ± 0.7	5.6 ± 3.2	0.07
Leptin (ng/mL)	1.4 ± 0.6	1.4 ± 0.2	0.91
Adiponectin (μg/mL)	21.8 ± 13.8	13.0 ± 2.9	0.33
Resistin (ng/mL)	41.5 ± 10.9	44.8 ± 2.9	0.63
FGF21 (pg/mL)	173.1 ± 60.9	628.8 ± 287.7	**0.01 ***
Osteocrin (pg/mL)	41.0 ± 23.2	41.9 ± 16.4	0.96
Osteonectin (ng/mL)	13.8 ± 12.4	13.7 ± 0.6	0.99

Serum adipocytokine and myokine levels in 24-month-old male WT (*n* = 5) and *Sirt7* KO (*n* = 3) mice after 16-h fasting. Data are expressed as the mean ± SD of each group. ** p* < 0.05. FGF21, fibroblast growth factor 21; GIP, glucose-dependent insulinotropic polypeptide; GLP-1, glucagon-like peptide-1; PAI-1, plasminogen activator inhibitor-1.

## Data Availability

Raw and processed RNA sequencing were deposited in the Gene Expression Omnibus (GEO) under accession number GSE207272. Any other relevant data are available from the corresponding author upon reasonable request.

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
