# Peer review of "SIRT7 Deficiency Protects against Aging-Associated Glucose Intolerance and Extends Lifespan in Male Mice"

_cells, 2022, doi:10.3390/cells11223609_

Round 1

Reviewer 1 Report

This is a well-written manuscript, with three main outcomes:

1. Sirt7 knock-out in male mice extends their lifespan by about 10%; no difference found in females.

2. Aged Sirt7-KO mice have about 3.5-times more serum Fgf21 than WT mice. The liver Fgf21 expression maintains high through life in Sirt7-KO mice but not WT mice.

3. Atf4 and other UPR ER stress genes maintained high expression in the liver in aged Sirt7-KO mice, but not WT mice.

The authors suggest, repeatedely through the manuscript, that increased Fgf21 levels likely produce/contribute to the extended lifespan of the Sirt7-KO males. But there is no experiment supporting this statement. A good control experiment would be to look at Fgf21 levels in females, since they have the same genotype (Sirt7-KO), but lack the life expansion phenotype. In Discussion the authors try to reason but essentially say they did not perform this type of control analysis (or any other). To my opinion this is the main limitation of the study.

My other comment is regarding the Figure 4, in which authors show biochemical analysis of the murine blood. There seems to be trends towards altered liver parameters (ALT, billirubin, LD) which end up being not significant, but that might be simply because the authors used rather small sample sizes (5 WT mice and only 3 KO mice). My concern is mainly the sample sizes in these analyses, abut also a lack of comments by the authors regarding the observed trends.

Another comments regarding Figure 4. The authors show that Sirt7-KO mice weigh less than WT, implying this is a weight gain/metabolic/body fat phenotype. My question is - how is the gross phenotype of the Sirt7-KO mice? Are these perhaps shorter/having skeletal phenotype that would contribute to the weight differences? How are the adipose tissue ammounts in Sirt7-KO mice?

Reviewer 2 Report

In the manuscript, Tomoya Mizumoto and co-workers report that male mice with SIRT7 deficiency extends lifespan because of an increase of fibroblast growth factor 21, but I didn’t find the direct evidence to support this conclusion. It seems odd why such a big difference between males and females. The authors should give the exact cause. In addition, what the mechanism of the increase of hepatic FGF21 expression? whether FGF21 is related to those cardiac phenotypes? Overall, more evidence is needed to support this conclusion.

Author Response

Response to the comments of Reviewer 2

We thank the reviewer for his/her constructive suggestions. We have substantially revised our manuscript based on his/her valuable comments.

In the manuscript, Tomoya Mizumoto and co-workers report that male mice with SIRT7 deficiency extends lifespan because of an increase of fibroblast growth factor 21, but I didn’t find the direct evidence to support this conclusion.

The reviewer is correct. We agree that FGF21 data for aged female Sirt7 KO mice are very important. However, these mice are not available now and thus the experiments would take several years to complete. Therefore, in accordance with the editor’s comments that “the manuscript needs restructuring and lowering of their claims,” we have revised a part of the Discussion section as follows: “there is a possibility that the increase of serum FGF21 levels might contribute to the extended lifespan of male Sirt7 KO mice. However, female Sirt7 KO mice did not show an extension of lifespan, but we did not examine their serum FGF21 levels. Thus, we cannot conclude that the prolonged lifespan of our male Sirt7 KO mice was due to increased FGF21 levels. It has been reported that female mice exhibit higher serum concentrations of FGF21 than males (Allard C, 2019). Investigation of serum FGF21 levels in female mice is a major future undertaking to address the contribution of FGF21 to the prolonged lifespan of male Sirt7 KO mice.” (lines 389 to 396). We have also revised the text (lines 37 to 38, and 74 to 75). We have changed the title of the revised manuscript to “SIRT7 deficiency protects against aging-associated glucose intolerance and extends lifespan in male mice” and deleted the original Figure 6 (schema). We also deleted the sentence “Thus, it is plausible that the increased serum levels of FGF21 could contribute to the lifespan extension and improved glucose tolerance observed in aged male Sirt7 KO mice.” (lines 312 to 314 of the original manuscript).

It seems odd why such a big difference between males and females. The authors should give the exact cause.

We thank the reviewer for pointing this out. It has been reported that sex differences affect the lifespan of mice (Ladiges W, 2009). For example, female IGF-1 receptor heterozygous KO (Igf1r+/−) mice live 33% longer than WT females, whereas there is no significant lifespan extension in male Igf1r+/− mice (Holzenberger M, 2003). Male, but not female, transgenic mice overexpressing Sirt6 have a significantly longer lifespan than WT mice (Kanfi Y, 2012). Female adipose tissue-specific extracellular nicotinamide phosphoribosyltransferase knock-in (ANKI) mice exhibit a significant extension of median lifespan, but male ANKI mice exhibit no such extension (Yoshida M, 2019). We agree that it is very important to clarify the reasons for these sex-based differences. However, it would be a major undertaking that lies outside the scope of this initial study.

In addition, what the mechanism of the increase of hepatic FGF21 expression? whether FGF21 is related to those cardiac phenotypes? Overall, more evidence is needed to support this conclusion.

ATF4 activates the transcription of Fgf21 by binding to its promoter (de Sousa-Coelho AL, 2012; Salminen A, 2017). The hepatic expression of Atf4 was significantly increased in aged male Sirt7 KO mice compared with WT mice (Figure 5F). Therefore, we think that the increased expression of hepatic Fgf21 and serum FGF21 levels in aged male Sirt7 KO mice might be, at least in part, a consequence of higher ATF4 expression (lines 426 to 432).

            It has been reported that FGF21 protects against cardiac hypertrophy and inflammation (Planavila A, 2015). Thus, higher FGF21 levels might be involved in the absence of cardiac dysfunction in our aged Sirt7 KO mice. However, the cardiac phenotypes were similar between aged Sirt7 KO mice (higher FGF21 levels) and WT mice (lower FGF21 levels). Thus, further studies are necessary to conclude the involvement of FGF21 in the cardiac phenotypes.

References

Allard, C. et al. Activation of Hepatic Estrogen Receptor-α Increases Energy Expenditure by Stimulating the Production of Fibroblast Growth Factor 21 in Female Mice. Mol Metab 2019, 22, 62–70, doi:10.1016/j.molmet.2019.02.002.

de Sousa-Coelho, A.L. et al. Activating Transcription Factor 4-Dependent Induction of FGF21 during Amino Acid Deprivation. Biochemical Journal 2012, 443, 165–171, doi:10.1042/BJ20111748.

.

Holzenberger M. et al. IGF-1 receptor regulates lifespan and resistance to oxidative stress in mice. Nature 2003, 421, 182-187. doi: 10.1038/nature01298.

Kanfi, Y. et al. The Sirtuin SIRT6 Regulates Lifespan in Male Mice. Nature 2012, 483, 218–221, doi:10.1038/nature10815.

Ladiges, W. et al. Lifespan Extension in Genetically Modified Mice. Aging Cell 2009, 8, 346–352. doi:10.1111/j.1474-9726.2009.00491.x

Planavila, A. et al. Fibroblast Growth Factor 21 Protects the Heart from Oxidative Stress. Cardiovasc Res 2015, 106, 19–31, doi:10.1093/cvr/cvu263.

Salminen, A. et al. Regulation of Longevity by FGF21: Interaction between Energy Metabolism and Stress Responses. Ageing Res Rev 2017, 37, 79–93. doi: 10.1016/j.arr.2017.05.004.

Yoshida, M. et al. Extracellular Vesicle-Contained ENAMPT Delays Aging and Extends Lifespan in Mice. Cell Metab 2019, 30, 329-342.e5, doi:10.1016/j.cmet.2019.05.015.

Reviewer 3 Report

This work by Mizumoto et al. expands the phenotype characterization of the SIRT7 deficient mouse modell. The authors provide very interesting and highly relevant data for the ageing field, once more pinpointing to the key role of several sirtuins, here SIRT7, and not only SIRT1, in lifespan regulation and ageing. The authors identify an sex (male)- and age-associated increase survival upon SIRT7 deficiency. This phenotype is associated with a better metabolic outcome as demonstrated by a better glucose tolerance of SIRT7 deficient animals. Serum profiling revealed more than 3.5-fold higher FGF21 levels in fasting SIRT7 KO mice as compared to the known decrease of FGF21 in wild-type control animals. Overall, the study is well done, described and especially discussed, given also the conflicting cardiovascular data of SIRT7 deficient mice in the literature and possible mouse strain incluence.

Major critics and recommendations:

1.       Provide the FGF21 data also for aged SIRT7 deficient female mice.

2.       I would increase the number of analysed male animals regarding the FGF21 serum levels, given this key point in the work.

3.       Regarding statistics, S.D. instead of S.E.M. should be used when handling small data n sizes (3-4).

Author Response

Response to the comments of Reviewer 3

We thank the reviewer for stating that our “study is well done, described and especially discussed” and for his/her constructive suggestions. We have substantially revised our manuscript thanks to his/her valuable comments.

This work by Mizumoto et al. expands the phenotype characterization of the SIRT7 deficient mouse modell. The authors provide very interesting and highly relevant data for the ageing field, once more pinpointing to the key role of several sirtuins, here SIRT7, and not only SIRT1, in lifespan regulation and ageing. The authors identify an sex (male)- and age-associated increase survival upon SIRT7 deficiency. This phenotype is associated with a better metabolic outcome as demonstrated by a better glucose tolerance of SIRT7 deficient animals. Serum profiling revealed more than 3.5-fold higher FGF21 levels in fasting SIRT7 KO mice as compared to the known decrease of FGF21 in wild-type control animals. Overall, the study is well done, described and especially discussed, given also the conflicting cardiovascular data of SIRT7 deficient mice in the literature and possible mouse strain incluence.

Major critics and recommendations: Provide the FGF21 data also for aged SIRT7 deficient female mice.

The reviewer is correct. We agree that FGF21 data for aged female Sirt7 KO mice are very important. However, these mice are not available now and thus the experiments would take several years to complete. Therefore, in accordance with the editor’s comment that “the manuscript needs restructuring and lowering of their claims,” we have revised a part of the Discussion section as follows: “there is a possibility that the increase of serum FGF21 levels might contribute to the extended lifespan of male Sirt7 KO mice. However, female Sirt7 KO mice did not show an extension of lifespan, but we did not examine their serum FGF21 levels. Thus, we cannot conclude that the prolonged lifespan of our male Sirt7 KO mice was due to increased FGF21 levels. It has been reported that female mice exhibit higher serum concentrations of FGF21 than males (Allard C, 2019). Investigation of serum FGF21 levels in female mice is a major future undertaking to address the contribution of FGF21 to the prolonged lifespan of male Sirt7 KO mice.” (lines 389 to 396). We have also revised the text (lines 37 to 38, and 74 to 75). We have changed the title of the revised manuscript to “SIRT7 deficiency protects against aging-associated glucose intolerance and extends lifespan in male mice” and deleted the original Figure 6 (schema). We also deleted the sentence “Thus, it is plausible that the increased serum levels of FGF21 could contribute to the lifespan extension and improved glucose tolerance observed in aged male Sirt7 KO mice.” (lines 312 to 314 of the original manuscript).

I would increase the number of analyzed male animals regarding the FGF21 serum levels, given this key point in the work.

We thank the reviewer for pointing this out. We have added the data of FGF21 levels of aged WT and Sirt7 KO mice in accordance with the reviewer’s suggestion (30-month-old WT mice, n = 6 in the original Figure 5A, n = 9 in the new Figure 5A; and 30-month-old Sirt7 KO mice, n = 5 in the original Figure 5A, n = 7 in the new Figure 5A).

Regarding statistics, SD instead of SEM should be used handling small data n sizes (3-4).

We thank the reviewer for pointing this out. We used SD instead of SEM in the revised manuscript. We have also increased the number of samples for qRT-PCR (new Figure 5B and 5F: 4-month-old WT mice, n = 9; 24-month-old WT mice, n = 7; 4-month-old Sirt7 KO mice, n = 7; and 24-month-old Sirt7 KO mice, n = 6).

References

Allard, C. et al. Activation of Hepatic Estrogen Receptor-α Increases Energy Expenditure by Stimulating the Production of Fibroblast Growth Factor 21 in Female Mice. Mol Metab 2019, 22, 62–70, doi:10.1016/j.molmet.2019.02.002.

Reviewer 4 Report

In this manuscript, Mizumoto, et al. reported their findings of SIRT7, a nicotinamide adenine 42 dinucleotide-dependent lysine deacetylases/deacylase, on lifespan in mice. They found that SIRT7 deficiency in mice extended the lifespan in male mice, but not in female mice. They further showed that SIRT7 knockout increased hepatic fibroblast growth factor 21 (FGF21) production and the serum level of FGF21 in male mice, which contributed to the phenotype of prolonged lifespan of male mice. Some of the findings are controversial to the literature. Additionally, the mechanism is largely unclear, though they claim that elevated FGF21 and the ATF4 UPR pathway in male aged SIRT7 KO mice. However, how SIRT7 regulate these pathways is not addressed. Moreover, they did not show such data in female mice, which makes it doubtful SIRT7 regulates longevity through UPR pathways and the FGF21. Major concerns are shown below.   

Major comments:

1. The phenotypes of SIRT7 KO mice are controversial to other reports in the literature, including the lifespan and cardiac phenotype. Though the authors mentioned the differences in the discussion (lines 406-418), the explanations are not convincing. In a previous study by Vakhurusheva, 2008 (Ref19 of this manuscript), the SIRT7 KO mice (the same mouse line used in this study, as confirmed by the authors) showed premature aging phenotypes, shortened lifespan, and hypertrophic cardiomyopathy. No data to show the SIRT7 deficiency in the mice they used. It has been reported that SIRT7 expression was decreased with aging, while increased during calorie restriction, indicating that SIRT7 most likely exhibits protective effect in longevity. Though such controversial findings are worth to be clarified, the current study did not provide sufficient and convincing data to verify the phenotypes of SIRT7 deficiency mice. Thus, the conclusion of this study (last paragraph of the discussion, line 419-424) was not appropriate and over-interpreted.

2.  SIRT7 deficiency extended the lifespan in male mice, but not in female mice. The authors claimed that elevated FGF21 by SIRT7 deficiency contributed to the phenotype in male mice. However, no FGF21 data from female mice was shown. This data is important, since without direct comparing the FGF21 changes in both male and female mice, it cannot conclude that the prolonged lifespan in male SIRT7 KO mice was due to increased FGF21 level.

3. Relative to above concern, this study did not address the mechanism on how SIRT7 regulate FGF21 production. Does the mechanism differ in male and female mice, and why?

4. Also, the data showed that ATF4 pathway is activated in the liver of aged male SIRT7 KO mice. Is ATF4 the direct target of SIRT7? Does SIRT7 deficiency affect the activity PERK kinase? In addition, dose ablation of SIRT7 also affect other UPR pathways, including the ATF6 and IRE1?

5. In figure2, how the mean lifespan and maximum were calculated? What’s the difference between mean and median lifespan? Based on the data of Figure 2A, the maximum lifespan looks similar in WT and SIRT7 KO mice.

Minor comments:

1. Some references are not cited properly. For example, line 44-46 cited Ref2. However, in this review paper, only the effect of Sirt2, but not all siturins, on longevity was summarized. More related reference(s) should be cited here.

2. How echocardiography was performed in awake mice? (line 93)

3. Unpaired Student’s t-test was used to compare the differences between two groups, but cannot be used compare differences in multiple groups, for example, Figure 5A, 5F and 5G.

Round 2

Reviewer 4 Report

The authors have provided explanations to address the comments, but did not provide the key data on the FGF21 levels in female mice. They explained that no aged female mice are available, but surprisingly there were no serum or tissues harvested from aged female mice. Though some of the major concerns were not well addressed, the authors have lowered their conclusion and the revised manuscript is improved compared with the original version. The manuscript can be accepted for publication with limitations of the current study (e.g. some key experiments were not conducted in female mice, etc.) should be added.